# Atmospheric Deposition around the Industrial Areas of Milazzo and Priolo Gargallo (Sicily–Italy)—Part A: Major Ions

**DOI:** 10.3390/ijerph20053898

**Published:** 2023-02-22

**Authors:** Filippo Brugnone, Walter D’Alessandro, Francesco Parello, Marcello Liotta, Sergio Bellomo, Vincenzo Prano, Lorenza Li Vigni, Mario Sprovieri, Sergio Calabrese

**Affiliations:** 1Dipartimento di Scienze della Terra e del Mare, Università degli Studi di Palermo, Via Archirafi, 36, 90123 Palermo, Italy; 2Istituto Nazionale di Geofisica e Vulcanologia, Section of Palermo Via Ugo la Malfa, 153, 90146 Palermo, Italy; 3Istituto per lo Studio degli Impatti Antropici e Sostenibilità in Ambiente Marino, Consiglio Nazionale delle Ricerche (IAS—CNR), Capo Granitola, Via del Mare, 3, Torretta Granitola, Fraz, 91021 Campobello di Mazara, Italy

**Keywords:** atmospheric deposition, acidity neutralization, major ions, marine source, anthropogenic source

## Abstract

The chemical composition of rainwater was studied in two highly-industrialised areas in Sicily (southern Italy), between June 2018 and July 2019. The study areas were characterised by large oil refining plants and other industrial hubs whose processes contribute to the release of large amounts of gaseous species that can affect the chemical composition of atmospheric deposition As in most of the Mediterranean area, rainwater acidity (ranging in the study area between 3.9 and 8.3) was buffered by the dissolution of abundant geogenic carbonate aerosol. In particular, calcium and magnesium cations showed the highest pH-neutralizing factor, with ~92% of the acidity brought by SO_4_^2−^ and NO_3_^−^ neutralized by alkaline dust. The lowest pH values were observed in samples collected after abundant rain periods, characterised by a less significant dry deposition of alkaline materials. Electrical Conductivity (ranging between 7 µS cm^−1^ and 396 µS cm^−1^) was inversely correlated with the amount of rainfall measured in the two areas. Concentrations of major ionic species followed the sequence Cl^−^ > Na^+^ > SO_4_^2−^ ≃ HCO_3_^−^ > ≃ Ca^2+^ > NO_3_^−^ > Mg^2+^ > K^+^ > F^−^. High loads of Na^+^ and Cl^−^ (with a calculated R^2^ = 0.99) reflected proximity to the sea. Calcium, potassium, and non-sea-salt magnesium had a prevalent crustal origin. Non-sea salt sulphate, nitrate, and fluoride can be attributed mainly to anthropogenic sources. Mt. Etna, during eruptive periods, may be also considered, on a regional scale, a significant source for fluoride, non-sea salt sulphate, and even chloride.

## 1. Introduction

The atmosphere is one of the geochemical spheres within which organic and inorganic chemicals, both from natural and anthropogenic sources, are injected, transported, and, at different times, removed through different physicochemical mechanism. The chemical composition of rainwater depends on (i) the chemical composition of gases and particles emitted by different sources, (ii) the emission and source amplitude, (iii) the chemical and physical reactions occurring during local and regional scale transport, and (iv) the removal processes [1,2]. Gaseous species and atmospheric aerosol particles include sea salts and crustal dust, which is one of the prominent aerosols in the atmosphere [3,4]. Volcanic emissions, biogenic material, and anthropogenic emissions are the main sources of chemical elements in rainwater. Meteorological conditions and distance from the coastline may also influence the relative contribution of each source [5,6,7].

Sodium, chloride, magnesium, and sulphate are major constituents of sea salt [8,9,10] and represent the main part of the total ions’ content of rainwater in coastal areas [11,12,13], although an enrichment of Cl^−^ may be attributed to anthropogenic sources, such as incinerators [14]. Calcium, but also magnesium, are typical crustal species [3], and especially non-sea-salt calcium is one of the main contributors to the neutralization of rainwater acidity [15]. Non-sea-salt sulphate and nitrate are usually considered mainly responsible for acidification processes and are the result of chemical reactions between anthropogenic emitted gaseous precursors SO_2_, and NO_x_ with oxidants, such as O_3_ and hydroxyl radicals (OH), that led to the formation of H_2_SO_4_ and HNO_3_ [3,16]. The latter may react with the particles of NaCl, releasing NaNO_3_ and HCl, and this reaction could explain the presence of NO_3_^−^ associated with coarse particles of Na^+^ from the marine origin [10]. These gaseous species come from anthropogenic sources such as energy production, urban transport, industry, and agriculture [17]. The deposition of ammonium (NH_4_^+^) mainly originates through ammonia volatilization from N-fertilization in agricultural fields and animal farming [3,18]. Ammonium is another chemical species that can contribute to the neutralization of rainwater acidity, but it may also trigger acidification of rainwater if oxidized to form nitric acid (HNO_3_). [19]. Gaseous and particulate fluoride (F^−^) are derived from anthropogenic emissions, such as coal burning, ceramic, brick, and cement factories [20], but also from natural sources, such as volcanoes [21] and seawater spray [22]. The abundance of potassium (K^+^) in rainwater could be attributed especially to biomass burning [23,24].

The long-range transport of air pollutants emitted by industrial activities from the European continent can influence the chemical composition of the rainwater in the Mediterranean basin. However, Saharan dust has been demonstrated to seasonally produce an impact on the chemical composition of rainwater in the Mediterranean area and sometimes even in northern Europe. Model estimates yield mineral dust deposition values from North Africa reaching 25–100 Tg yr^−1^ to the Mediterranean Basin [25]. Ions such as NH_4_^+^, NO_3_^−^, and SO_4_^2−^, are usually indicators of secondary aerosol and long-range transport [8]. Air mass trajectories combined with the chemical composition of rainwater provide information on the origin of atmospheric pollutants [26,27].

Long-term variations in the chemical composition of rainwater provide important footprints of the temporal evolution of atmospheric pollution and can be used to discriminate natural from anthropogenic sources [5]. Rainwater chemistry also provides a general picture of changes in the atmospheric composition caused by anthropogenic activities, as well as changes in meteorology and climate [28]. The same approach can be used to explore global element cycling in the atmosphere [29,30,31] and air mass circulation [30,31].

The European Environment Agency reported the existence of hundreds of thousands of contaminated sites in Europe [32], many of them resulting from earlier industrialization and poor environmental management. The dispersion and accumulation of contaminants, mainly chemicals, may derive from past and present industrial activities, and the subsequent pollution of air, groundwater, surface water, soil, and crops may affect human health [33].

The monitored areas are characterised also by a high density of vehicular traffic, both urban and freight-related, especially along the roads bordering the large industrial plants of the two AERCAs. Epidemiological information reported that the overall mortality in the population of the site appears to be in line with the regional average. Mortality from diseases of the digestive system was in excess in both genders, as it was for diseases of the urinary tract. Mesothelioma of the pleura resulted in excess, as a cause of death, in both genders. There was an excess in lung cancer mortality among women and acute respiratory diseases among men [34,35].

This paper illustrates the chemical composition of major ions in rainwater from two study areas located close to large industrial plants in southern Italy (Milazzo and Priolo Gargallo). Possible sources of different ion species are identified and characterised, describing the processes that determine the physicochemical characteristics (pH and Electrical Conductivity) of rainwater and the quantification of the atmospheric bulk depositions of major ions.

## 2. Study Area and Climatic Setting

The study areas located in Sicily (southern Italy) were classified by the Italian Ministry of the Environment as areas at high risk of environmental crisis (hereafter AERCA—Area ad Elevato Rischio di Crisi Ambientale). These sites were defined as highly contaminated areas where industrial pollution may cause an important impact in terms of human health and ecological risk, as well as a detrimental impact on cultural and environmental heritage [36].

Milazzo has about 30,000 inhabitants; it is placed on the northern coast of Sicily. Industrial development abruptly increased during the 1950s and 1960s, with a steel industry, energy production plants, and the largest single oil refinery in Europe, which produces low-sulphur diesel and unleaded gasoline [34,37], and which are still in operation. The industrial area covers a surface of about 5.5 km^2^ (Figure 1). Epidemiological information reported an increasing number of cancers and lung infections in this region [38], as well as an excess of diseases in the urinary system [39].

The AERCA of Priolo Gargallo covers an area of 55 km^2^ and comprises the municipalities of Priolo Gargallo (12,000 inhabitants), Augusta (36,000 inhabitants), Melilli (13,000 inhabitants), Siracusa (122,000 inhabitants), Floridia (23,000 inhabitants), and Solarino (8000 inhabitants). This AERCA includes a huge petrochemical district (the widest in Europe, about 27 km^2^) that extends from the north of Siracusa to Augusta, where the main industrial activities include oil refineries, processing oil-related products, and energy production.

Both study areas are characterised by Mediterranean climate conditions. During summer, subtropical high-pressure cells dominate, while, during winter, the main components of the weather conditions are cyclonic storms [40,41]. Data from four automatic weather stations belonging to the regional monitoring network (Servizio Informativo Agrometeorologico Siciliano—SIAS; http://www.sias.regione.sicilia.it/, accessed on 21 October 2022) have been acquired to determine the direction of the prevailing winds close to the ground and to estimate how long the bulk collectors were downwind with respect to the industrial plants. In addition, data on rainfall were used to inter-calibrate the amount of rainfall collected by the bulk collectors.

The automatic weather stations (yellow squares in Figure 1), whose data were taken from SIAS, were Torregrotta (26 m a.s.l.) for the Milazzo AERCA and Augusta (90 m a.s.l.) and Siracusa (90 m a.s.l.) for the Priolo Gargallo AERCA, while Palazzolo Acreide (640 m a.s.l.) was considered as local background. Meteorological parameters included air temperature (°C), relative humidity (%), precipitation (mm), average wind speed 2 m above the ground (m s^−1^), and dominant wind direction 2 m above the ground (°). Data on prevailing wind direction are shown in Figure 1, while the other meteorological parameters are included in Figure 2.

During the study period, the rainfall was evenly distributed at Milazzo, while it was more concentrated in the autumn period (September–November) at Priolo Gargallo. In the area of Milazzo, the winds blew from the Southeast (135°), at Palazzolo Acreide they blew from two dominant directions, North (0°) and Southwest (225°), while at Siracusa winds had two dominant directions, North (°) and West (270°). The weather station of Augusta recorded winds blowing mainly from the East (90°). These differences in wind directions were related both to the general circulation in the troposphere at the various monitoring sites and to local wind deflection effects caused by topography. For about 19% of the monitoring period at Milazzo, winds came from the W and NW. Consequently, the monitoring sites were the direction of the dispersion of industrial emissions. Monitoring sites in the Priolo Gargallo area were found to be exposed to the dispersion direction of industrial emissions for about 10% of the study period.

## 3. Materials and Methods

Monthly rainwater samples were collected from the two study areas over a year from June 2018 to July 2019, using a network of 11 bulk collectors (Table 1). A total of 131 samples were collected from June 2018 to July 2019, 60 from the Milazzo AERCA and 71 from Priolo Gargallo AERCA.

Sampling procedures followed published protocols for atmospheric composition monitoring [42,43,44,45]. Each bulk collector consisted of a 5 L bottle (made of High-Density Polyethylene—HDPE) and a Büchner type funnel (Ø 240 mm, made of polypropylene—PP) (for more details see Calabrese et al. 2011) [42]. Each sampling bottle was inserted in a 1.5 m high PVC tube to protect the collected rain from direct sunlight, and each collector was open during the entire exposure period. The bulk collectors were located on the roof of public or private buildings and two air quality monitoring stations of the Regional Agency for Environmental Protection (Agenzia Regionale per la Protezione Ambientale—ARPA Sicilia). To evaluate the impact of industrial pollution on the studied areas, five monitoring sites were located near the industrial area of Milazzo, and five closes to the industrial area of Priolo Gargallo. One additional bulk collector was placed at Palazzolo Acreide, about 30 km away from the AERCA of Priolo Gargallo (Figure 1 and Table 1), to estimate the chemical characteristics of the rainwater in an area not directly influenced by the chemical plants (local background). All the equipment of the bulk collectors (funnels and bottles) was cleaned before the field exposure, using “ultrapure” deionized water (18.2 MΩ cm resistance Milli-Q water-purification system, Millipore), and dried under a laminar hood for 24 h. Afterward, the bulk collectors were assembled and stored in clean plastic bags until their exposure in the field. In all monitoring campaigns, bottles and funnels were removed and replaced by clean ones at the end of the monthly sampling period. The samples were carried to the laboratory and the quantity of collected water was measured gravimetrically to calculate the amount of rainwater. An aliquot was used to measure Electrical Conductivity (EC) and pH by using a specific Orion conductivity meter and a WVR pH meter with a 52–21 combination electrode. The pH meter was calibrated with buffer solutions (pH 4, 7, and 10), and the conductivity meter was calibrated with a solution of 84 μS cm^−1^ @20 °C, similar to the average conductivity of the sampled rainwater. The other three aliquots were split for different analytical determinations: (i) unfiltered water was used for total alkalinity determinations through an automatic titrator (Titrator Compact G20—Mettler Toledo), using 0.01N HCl, and expressed as mg(HCO_3_^−^) L^−1^; (ii) filtered (0.45 µm) aliquot for major anions (NO_3_^−^, SO_4_^2−^, Cl^−^, and F^−^) determination through an ion chromatography system (ICS-1100 Dionex) in suppressed mode and equipped with an anion column (AS14A) and a pre-column (AG14A) that works under a continuous flow of carbonate-bicarbonate eluent; (iii) filtered and acidified (Ultrapure HNO_3_) sample analysed for major cations (Ca^2+^, K^+^, Mg^2+^, and Na^+^), through an ion chromatography system (ICS-1100 Dionex) in suppressed mode and equipped with a cation column (CS12A) and pre-column (CG12A) that works under a continuous flow of methane-sulfonic acid with eluent regeneration. Calibration solutions for all the investigated ions were prepared diluting stock standard solutions (Merck). Calibration curves were made with 6 calibration levels and precision was always better than 2% for cations and 3% for anions, except fluoride (11%). The was accuracy of the method was checked by analysing certified reference materials of natural waters (Environment Canada Ontario 99 and NWLON-7) at regular intervals. Sample manipulation and analysis were carried out in clean rooms, all plastic ware was washed with “ultrapure” deionized water, and all the aliquots were stored at 4 °C before the analytical determinations. Pre-treatment of the samples and chemical analyses were performed in the laboratories of the Dipartimento di Scienze della Terra e del Mare (DiSTeM), and the laboratory of the Istituto Nazionale di Geofisica e Vulcanologia (INGV), all situated in Palermo.

### 3.1. Analysis of Chemical Data

The statistics on pH, EC, and major ions concentrations (meq L^−1^) values are given in Table 2 (for the entire dataset see Brugnone et al., 2023 [46]). The annual volume-weighted mean (VWM) concentration values were calculated by using the following formula:
(1)$$\mathrm{VWM}=\frac{\sum_{i=1}^{n}CiPi}{\sum_{i=1}^{n}Pi}$$
where *Ci* represents the rainwater concentrations (mg L^−1^ for major ions) value in the *i*th sample, *Pi* is the rainfall depth (L m^−2^) during the *i*th sampling period and *n* is the total number of rainfall events [47]. This parameter was used to reduce the potential effect of different precipitation amounts on pH, Electrical Conductivity, and concentrations of water-soluble inorganic ions.

To estimate the contribution of the marine source and anthropogenic sources toward the different major ionic species, the non-sea-salt fractions ([X]nss) were calculated by the following Formula (2), and are also shown in Table 2:
(2)$$\left[X]\mathrm{nss}=\left[X\right]{\mathrm{rain}\;-\;([\mathrm{Na}}^{+}]\mathrm{rain}\;\times \;\left[\frac{X}{{\mathrm{Na}}^{+}}\right]\mathrm{SW}\right)$$
where [X]rain is the concentration of species X in rainwater, and [X]nss is the non-sea salt contribution of species X in meq L^−1^ [48].

According to Conradie et al. 2016 [49], the marine relative contribution to the total ion composition of the atmospheric deposition can be estimated by the formula:
(3)$$[X]\mathrm{marine}=\left[{\mathrm{Na}}^{+}\right]\mathrm{rain}\;\times \;\left[\frac{X}{{\mathrm{Na}}^{+}}\right]\mathrm{SW}$$
where [X]marine is the sea salt contribution of X in meq L^−1^, [Na^+^]rain is the concentration of Na^+^ in rain (meq L^−1^), and [X/Na^+^]SW is the seawater equivalent ratio [9,49].

The use of the previous equations assumes that all the sodium in rainwater comes from the sea and that no chemical fractionation occurs between chloride and the other ions after sea salt injection into the atmosphere [1].

Correlation indexes between major ions are shown in Table 3.

Annual atmospheric depositions (g m^−2^ yr^−1^), with relative uncertainty, are reported in Table 4 and were calculated by multiplying the annual VWM ionic concentrations by the total annual rainfall depth at each monitoring site.

According to the electroneutrality principle, the sum of positive and negative charges within the water sample should be zero. The charge balance error of each sample was calculated according to the equation:
(4)$$\mathrm{Balance}\;\mathrm{error}\;(\%)=\{[(\frac{\sum \;\mathrm{anions}\;}{\sum \mathrm{cations}})-1]\;\times \;100\}$$

The neutralization factors of the different chemical components of interest were estimated using the equation suggested by Possanzini et al. 1988 [50]:
(5)$$\mathrm{NFx}=\frac{\left[X\right]}{\left[\mathrm{nss}-{\mathrm{SO}}_{4}^{2-}+{\;\mathrm{NO}}_{3}^{-}\right]}$$
where [X] denotes the chemical component of interest, i.e., nss-Ca^2+^, nss-Mg^2+^, etc.

## 4. Results and Discussion

### 4.1. Chemical Composition

The charge balance error of each sample was calculated by using Equation (4) in Section 3.1. The resulting values were between −58% to 163%, with an arithmetical average value of 14%. In 33.6% of the cases (44/131 samples), the calculated error was between −10% to +10%, while in 57.2% of the cases (75/131 samples) the calculated error was between −20% to +20% (Figure 3).

Even though most of the analysed samples fall within the acceptable range (±0.2 meq L^−1^) for dilute waters [52], 53% of the samples displayed cation excess, while 47% of the samples displayed anion excess (Figure 3). Such a deficiency in anions might be attributed to anionic species, such as CH_3_COO^−^, HCOO^−^, C_2_O_4_^2−^, and PO_4_^3−^, which were not determined in this study, while a cation excess may be justified by the presence of undetermined NH_4_^+^.

The chemical composition in the two study areas had a similar pattern, with prevailing ions Ca^2+^, Na^+^, Cl^−^, intermediate concentrations of total alkalinity as HCO_3_^−^, SO_4_^2−^, and Mg^2+^, and lower concentrations of NO_3_^−^, K^+^ and F^−^ (Figure 4). Table 2 shows that concentrations (in meq L^−1^) of the major ionic species follow the sequence Cl^−^ > Na^+^ ≃ Ca^2+^ > SO_4_^2−^ ≃ HCO_3_^−^ > NO_3_^−^ ≃ Mg ^2+^ > K^+^ > F^−^.

Chloride was the most abundant anion, with a relative abundance with respect to all anions of 61.4% at Milazzo, 55.1% at Priolo Gargallo, and 59.8% at Palazzolo Acreide. The least abundant anion was F^−^ with a relative abundance of 0.6% in the Milazzo study area, 1.0% in the Priolo Gargallo study area, and 1.8% in the Palazzolo Acreide study site. Anthropogenic-related ions (nss-SO_4_^2−^ and NO_3_^−^) accounted altogether for 19.8 % considering the arithmetical average of the studied sites.

As for the cations, Na^+^ was the most abundant cation, with a relative abundance of 48.7% in the area of Milazzo, 42.7% in the area of Priolo Gargallo, and 37.3% in the study site of Palazzolo Acreide. The least abundant cation was K^+^ with a relative abundance of 2.9%, 2.8%, and 2.4% in the Milazzo, Priolo Gargallo, and Palazzolo Acreide study areas, respectively (Figure 4).

The pH values of the rainwater showed a high variability, ranging between 4.2 to 8.3 (median 6.7), and between 3.9 to 8.3 (median 6.3), in the Priolo Gargallo and Milazzo areas, respectively. The pH at Palazzolo Acreide varied between 4.8 and 7.3 (median 6.7). The frequency distributions of the pH are shown in Figure 5a. Most of the samples (81.4% at Priolo Gargallo, 78.0% at Milazzo, and 92.0% at Palazzolo Acreide) had a pH value greater than 5.6, showing a prevailing process of neutralization of rain acidity. The higher frequency of alkaline rainwater depends, therefore, on the high loading of cations in the air masses reaching the site. Previous authors evidenced widespread acid-rain neutralization by atmospheric carbonate dust over the entire Mediterranean basin induced by the frequent arrival of Saharan dust [2,25,52,53] and widespread outcrops of limestone formations [1,54]; the latter is specific evidence to explain the results achieved in our study areas. From this perspective, the observed acidity of some samples (18.6%, 22%, and 8% at Priolo Gargallo, Milazzo, and Palazzolo Acreide, respectively), was due to the influence of anthropogenic (urban and/or industrial) NO_x_ and SO_x_ gaseous pollutants, deriving from a combination of local and regional sources [55]. Halogen acids (HCl, HBr, and HF) from volcanic sources and organic acid produced by biomass burning may also contribute to the lowering of the pH of rainwater [12,21].

The wash-out of below-cloud alkaline aerosols neutralizes rain acidity deriving from anthropogenic sources, in agreement with previous information from the Mediterranean area [1,56].

In the present study, an average neutralization ratio of the acidity of rainwater of about 0.076 was estimated by using Equation (5) in Section 3.1. This value suggests that 92.4% of the acidity due to SO_2_ and NO_X_ has been neutralized by the alkaline dust. The neutralization factors of different major ions were estimated using the same equation for different major ions. Values obtained from the VWM of the study sites ranged from 1.03 to 1.96 for nss-Ca^2+^, from 0.39 to 0.90 for nss-Mg^2+^, and from 0.07 to 0.10 for nss-K^+^, suggesting that nss-Ca^2+^ was, permanently, the major neutralizing agent in the area. For nss-Ca^2+^, a seasonal variability in the magnitude of the NF was observed, with the maximum effect observed during periods with frequent high-intensity winds from southern directions and a reduced number of precipitation events. The main contribution of nss-Ca^2+^ to rainwater, at least in the Mediterranean basin, originated from carbonate rocks and soils. The study areas, as well as much of the territory of Sicily, are characterized by a vast coverage of rocks and soils rich in carbonate minerals. During periods characterized by low rainfall, soils with reduced moisture levels are affected by strong erosive processes, particularly if high-intensity winds concur with dry periods. Nonetheless, relevant soil erosion also occurs when high-intensity rainfall follows dry periods. Figure 6 shows the correlation between pH and nss-Ca^2+^, nss-SO_4_^2−^, and between nss-Ca^2+^ and nss-SO_4_^2−^ in all the study areas. Rainwater samples characterized by the high load of nss-SO_4_^2−^, for which we could expect relatively low pH values, also show a high load of nss-Ca^2+^, thus suggesting that the residual acidity from SO_2_-derived SO_4_^2−^ was neutralized by the high load of nss-Ca^2+^; differently, low concentrations of nss-Ca^2+^ correspond to low pH values because of limited buffering effects. The good positive correlation between nss-Ca^2+^ and nss-SO_4_^2−^, with an R^2^ value of 0.79, confirms the importance of crustal (carbonate) material in the rain neutralization process in both the study areas.

The lowest pH values were observed in all those samples collected after abundant rains, as the result of the lower influence of the dry deposition of alkaline materials. This observation is expected because, after abundant rain events, the dust is washed-out from the atmosphere lowering the content of alkaline ions in rainwater. Indeed, in the samples collected for this research, the values of pH were inversely related to the amount of precipitation (Figure 5b,c).

The Electrical Conductivity (EC) values, on a logarithmic basis, showed a high variability ranging from 1.08 to 2.60 (corresponding to values from 10 μS cm^−1^ to 317 μS cm^−1^), with a geometric average value of 1.65 (61 μS cm^−1^), and from 1.10 to 2.50 (12–396 μS cm^−1^), with a geometric average value of 1.69 (71 μS cm^−1^) in the Milazzo and Priolo Gargallo study area, respectively. As regards the background site of Palazzolo Acreide, the EC varied between 0.86 and 1.73 (7–53 μS cm^−1^), with a geometric average value of 1.43 (31 μS cm^−1^), lower than the values observed in the sites closer to anthropogenic emissions and to the sea. EC values were generally very low in all areas, with values below 100 μS cm^−1^ for 84% of the samples (Figure 7a), and they were, indeed, generally inversely correlated with the amount of rainwater (Figure 7b,c). These EC values are typical of rain with a relatively low amount of dissolved ions.

### 4.2. Sources and Processes

Chloride and sodium showed high concentrations at the coastal monitoring sites in both the Priolo Gargallo and Milazzo areas, with median values of 6.69 mg L^−1^ and 7.00 mg L^−1^ for chloride and 3.6 mg L^−1^ and 3.27 mg L^−1^ for sodium in the two study areas, respectively. The concentrations of these two ionic species were lower in Palazzolo Acreide where, due to the greater distance from the coastline, they were less affected by the marine source (median 2.26 mg L^−1^ and 1.30 mg L^−1^ for chloride and sodium, respectively) (Figure 8a,b).

Chloride and sodium are generally used as tracers of marine origin if the Na^+^/Cl^−^ ratio in rainwater does not differ significantly from that of seawater (0.86 on an equivalent basis [9]). Chloride enrichments are mostly related to human activities such as the burning of solid waste [9], while sodium excess can be attributed to a terrigenous component or human activities such as emissions from cement factories or detergent factories. The Na^+^/Cl^−^ ratios in rainwater in all the sampling sites were close to the seawater ratio (0.76) and the high correlation in both study areas (R^2^ 0.98 and 0.99 for Milazzo and Priolo Gargallo, respectively), denoting a pronounced regional contribution of marine aerosols and a negligible influence of human activities or crustal dust (Figure 9 and Table 3).

The rainwater samples with Cl^−^ excess were more than those having Na^+^ excess. Since Sicily is an island surrounded by the Mediterranean Sea, the sea has always been a major source of Na^+^ and Cl^−^ (from 84% to 100% of the total concentrations). As for the absolute concentrations of chloride, the obtained marine fractions calculated from the VWM decreased from the coast to the inner study area. The Palazzolo Acreide site was that at which the lowest concentrations of Na^+^ and Cl^−^ were recorded. The elements which were most influenced by sea salt contribution were Cl^−^ (from 84% to 100%) and Mg^2+^ (from 66% to 90%). The Mg^2+^/Na^+^ ratio, equal to0.289 on equivalent basis, was also close to that of seawater (0.224), with strong correlations observed between Na^+^ and Mg^2+^ (R^2^ = 0.96), (Figure 9b and Table 3), and Cl^−^ and Mg^2+^ (R^2^ = 0.95), (Table 3), in all sampling sites. This indicates that the Mediterranean Sea is also a strong source of Mg^2+^ in the Sicily region. The highest median concentrations of magnesium were measured in the Priolo Gargallo area (0.64 mg L^−1^), while the lowest concentrations were recorded in Palazzolo Acreide, with median values of 0.34 mg L^−1^ (Figure 8b).

The highest median concentration of potassium was measured in the Priolo Gargallo area (0.29 mg L^−1^), while the lowest concentration was recorded in Palazzolo Acreide (0.17 mg L^−1^ (Figure 8b). Total sulphate concentrations were comparable between the Priolo Gargallo area (median 3.78 mg L^−1^) and the Milazzo area (median 3.65 mg L^−1^), while lower concentrations (median 2.31 mg L^−1^) were recorded in Palazzolo Acreide (Figure 8a). Sulphate and potassium displayed high variable contributions from sea salt (from 20% to 50% and from 33% to 90%, respectively). The greatest influence by sea spray, with a value of 78% of the total ions VWM concentration, was recorded at the Milazzo site, located at 0.36 km from the coastline, at an elevation of 20 m a.s.l., and exposed, seasonally, to sea breeze winds. Moving away from the coast and rising in elevation, the contribution from the sea spray gradually decreases, reaching the lowest values (55%) at Palazzolo Acreide (Priolo Gargallo), the local background site located 28.2 km from the coastline and at an elevation of 642 m a.s.l., and in the site of Melilli, (Priolo Gargallo), located 5.15 km from the coastline and an elevation of 249 m a.s.l., with a value of 54% of the total ions VWM concentration.

Differentiating between sea-salt sulphate (ss-SO_4_^2−^) and non-sea salt sulphate (nss-SO_4_^2−^), it was observed that, in the Priolo Gargallo area, there was a greater contribution of the latter than in the Milazzo area, a direct consequence of the higher SO_x_ emissions from the industrial hub. The marine source contributed to the sulphate in coastal monitoring sites while its contribution was almost negligible at the Palazzolo Acreide site where a median concentration of ss-SO_4_^2−^ of 0.49 mg L^−1^ has been calculated (Figure 8a).

Nitrate showed comparable concentrations in the two heavily urbanised areas of Milazzo and Priolo Gargallo (median 1.65 mg L^−1^ and 1.54 mg L^−1^), while the concentrations were lower (median 1.21 mg L^−1^) in the small town of Palazzolo Acreide (Figure 8a).

Sulphate and nitrate mainly originate from anthropogenic sources, and they are also indicators of secondary aerosol formation and long-range transport [8,57,58]. Coal burning, ceramic industries, and cement production are the main human activities that emit fluoride into the atmosphere [59]. Relevant sources of sulphate, fluoride, and chloride in Sicily are volcanic emissions, mainly from Mount Etna and, to a lesser extent, from Stromboli and Vulcano [20,60,61].

Non-sea-salt sulphate can be attributed mainly to local fossil fuel combustion sources, but also to the long-range transport from the most heavily industrialized parts of Europe. It results from emissions of gaseous sulphur dioxide (SO_2_) during the combustion of sulphur-containing fuels (oil, coal, and diesel) and sulphide ore smelting. In the atmosphere, gaseous SO_2_ is oxidised to sulphate. Both the Milazzo and Priolo Gargallo AERCA have important sources of anthropogenic sulphur species (SO_x_). Based on the ARPA Sicilia 2022 Report on atmospheric emissions of gases, particulates and metals, the annual emissions (2015, [62]) of SO_x_ were 3728 tons in the Milazzo AERCA (Figure 10a) and 12,305 tons in the Priolo Gargallo AERCA (Figure 10b). Of the SO_x_ emissions, 98.5% were attributable to industrial sources, specifically, 53% and 47% to combustion processes used in the electric power industry and other industrial processes that do not involve fossil fuel combustion, respectively. The remaining 1.5% of SO_x_ emissions came from non-local and/or mobile sources.

Nitrate is the final product of the multiphase reaction processes of gaseous NO_x_, atmospheric particles, and cloud water [2,59]. Gaseous NO_x_ is primarily emitted as nitrogen oxide (NO) from natural (such as biomass burning), as well as anthropogenic sources (vehicular traffic, household combustion, and fossil fuel combustion). The proximity of the monitoring sites to major industrial clusters and their location in densely urbanized areas makes it possible to assume that almost all NO_3_^−^ may be of anthropogenic origin. Based on the ARPA Sicilia 2022 Report, the annual emissions (2015) of NO_x_ were 3675 tons in the Milazzo AERCA (Figure 10a) and 9836 tons in the Priolo Gargallo AERCA (Figure 10b). Of the NO_x_ emissions, 81.5% (the average of the two study areas) were attributable to industrial sources, specifically, 82% and 18% to combustion of fossil fuel and other industrial processes that do not involve fossil fuel combustion, respectively. The remaining 18.5% of NO_x_ comes from emissions from vehicles.

The positive correlation between nitrate and nss-sulphate (R^2^ = 0.47 and 0.89 in Milazzo and Priolo Gargallo areas, respectively), (Figure 9c and Table 3), corroborates the hypothesis that nitrate can be partially attributed to the same fossil fuel combustion sources as for nss-sulphate, and the rest to other sources, such as vehicular traffic. The ratio between nss-SO_x_ and NO_x_ may be useful in identifying pollution sources, because fuels used for electricity generation and transportation differ in their sulphur content and because the ratio is related to combustion conditions. Typically, electricity production is expected to result in a lower nss-SO_x_/NO_x_ ratio than emissions caused by low-temperature boilers burning fuel oil with high sulphur content [60]. Previous work [60] has pointed out that mobile (vehicular traffic) and point sources (refineries and power plants) may be identified by their characteristic nss-SO_x_/NO_x_ ratio which is, therefore, a useful indicator of pollution sources. The ratio is generally greater than 1 in industrialized areas, while lower than 0.5 in areas located away from large stationary sources. On an equivalent basis, annual median ratios between 1.3 and 1.7 and from 1.4 to 2.0 were calculated for Priolo Gargallo and Milazzo study areas, respectively (see also Figure 9c). The lowest values were calculated for Melilli and Palazzolo Acreide (1.3), and the highest values for Siracusa (1.7) and Milazzo (2.0). The greater distance from the industries and the higher elevation with respect to those placed approximately at sea level is the reason why the lowest ratios were observed in Melilli and Palazzolo Acreide. However, the same ratios are greater than 1, probably due to the low NOx emissions from vehicular traffic, as they are medium towns of about 13,000 and 8000 inhabitants, respectively. The highest values were recorded at two monitoring sites highly exposed to SOx emissions from fossil fuel combustion processes and power generation plants in the two industrial hubs of Priolo Gargallo and Milazzo.

High calcium concentrations were recorded at all monitoring sites and, especially, in the Priolo Gargallo area, which is characterised by extensive outcrops of rocks with carbonate composition, and is that most affected by Saharan dust transport and fallout during sirocco episodes. The median concentrations were 3.56 mg L^−1^, 1.97 mg L^−1,^ and 1.56 mg L^−1^ for the study areas of Priolo Gargallo, Milazzo, and Palazzolo Acreide respectively (Figure 8b). Calcium showed very low sea salt contribution (from 4% to 11%) and mostly derived from the weathering of the carbonate rocks which crop out extensively in the Sicily region. The terrigenous (crustal) contribution to the chemical composition of bulk deposition is linked also to Saharan dust species, e.g., calcite, dolomite, gypsum, illite, smectite, and palygorskite. Seasonally, several tons of dust particles are carried by southerly winds from the Sahara Desert reaching the entire Mediterranean basin and sometimes much of the European continent.

Fluoride concentrations were generally low in all the monitoring sites, with the highest median values measured in the Milazzo area (0.046 mg L^−1^) and the lowest at Palazzolo Acreide (0.036 mg L^−1^). However, peaks of concentrations (up to 0.88 mg L^−1^) were recorded in the Priolo Gargallo area in December 2018, coinciding with an Etna eruptive event (Figure 8a). Gaseous fluoride compounds may derive from anthropogenic sources, but volcanic emissions from Mt. Etna represent an important natural source of this element [21,60,61]. Mount Etna discharges a permanent volcanic plume consisting of magmatic gases such as CO_2_, SO_2_, HCl, and HF, with much smaller amounts of particulate sulphates, chlorides, and fluorides [63]. Both these sources emit fluoride in a very soluble form (HF), allowing a rapid removal from the atmosphere by wet deposition close to the emission points [61]. Longer dry spells may favour its travel to sampling sites further away from Mt. Etna. A marine origin of F^−^ can be excluded due to the very low correlation with Na^+^ (Table 3), and Cl^−^ (Figure 9d), and therefore the Na/F and Cl/F ratios were very different from those for rainwater, in which the main source of fluoride is seawater. The best correlation between fluoride and other ions was with nss-sulphate (R^2^ = 0.40), (Table 3), which may derive both from volcanic and anthropogenic SO_2_ emissions.

### 4.3. Atmospheric Deposition Fluxes for Major Ions

From concentrations (g L^−1^), exposure time (yr^−1^) of the collectors, and amount of rainfall (L m^−2^), the annual atmospheric deposition fluxes (g m^−2^ yr^−1^) for each major ion at each sampling site were calculated and reported in Table 4. Atmospheric deposition values followed the concentrations of various ion species but also depend on rainfall amounts. Fluoride depositions were very low (0.06 ± 0.003 g m^−2^ yr^−1^ to 0.09 ± 0.014 g m^−2^ yr^−1^) and almost homogeneous in the two study areas. Chloride and sodium had decreasing deposition values moving from coastal monitoring sites to those located further from the coast, because of the predominantly marine source of these two ions. The values were between 9.84 ± 0.65 g m^−2^ yr^−1^ and 32.25 ± 3.18 g m^−2^ yr^−1^ for chloride and between 4.57 ± 0.34 g m^−2^ yr^−1^ and 15.42 ± 1.52 g m^−2^ yr^−1^ for sodium. For nitrate, the highest values were calculated for Melilli (3.06 ± 0.18 g m^−2^ yr^−1^) and Passo Vela (2.60 ± 0.29 g m^−2^ yr^−1^) for the Priolo Gargallo AERCA and the Milazzo AERCA, respectively, while for sulphate, the highest values were found at the two monitoring sites of Siracusa (up to 7.77 ± 0.55 g m^−2^ yr^−1^) and Milazzo (up to 7.81 ± 0.45 g m^−2^ yr^−1^), i.e., at the sites most exposed to SO_x_ emissions from industrial plants located in their vicinity. The lack of correspondence between the sites characterised by the greatest nitrate depositions and those with the greatest sulphate depositions was mainly due to the different emission ratios of these two species in the two study areas. The median depositions of potassium (0.56 ± 0.04 g m^−2^ yr^−1^), magnesium (1.18 ± 0.12 g m^−2^ yr^−1^), and calcium (4.63 ± 0.30 g m^−2^ yr^−1^) were homogeneous among the various monitoring sites and reflected the common geogenic source, on a regional scale, of these ionic species.

Investigating in detail the monthly distribution of atmospheric depositions of selected major elements, it was observed that sulphate shows little variation throughout the year, with high monthly values (up to 528 mg m^−2^) and comparable median values at the two industrial sites (163 mg m^−2^ for Priolo Gargallo AERCA and 182 mg m^−2^ for Milazzo AERCA) (Figure 11a and Figure 12a). A clear seasonal distribution was observed for chloride. Having recognised the sea as the main source of chloride in the rainfall sampled in the different study areas, its transport and subsequent deposition on land were strongly influenced by meteorological parameters such as wind direction and intensity and rainfall amounts. It is therefore not surprising that the lowest monthly deposition values were recorded during the least rainy months characterised by high atmospheric stability and light winds, while the highest values of up to 2905 mg m^−2^ for Priolo Gargallo AERCA and up to 3355 mg m^−2^ for Milazzo AERCA were recorded during the rainiest months with the greatest atmospheric instability and strong winds. Comparable median monthly atmospheric deposition values were also found for chloride in the two areas, equal to 286 mg m^−2^ for the Priolo Gargallo AERCA and 299 mg m^−2^ for the Milazzo AERCA (Figure 11b and Figure 12b).

Monthly fluoride deposition was consistently very low in both study areas, although a higher median was observed for the Milazzo AERCA (1.8 mg m^−2^) than for the Priolo Gargallo AERCA (1.1 mg m^−2^), which may be explained by the presence of larger anthropogenic sources of fluoride (brick factories and bigger oil refining plants) in the former industrial area than in the latter. At the end of December 2018, following an intense eruptive phase of Mt. Etna, the volcanic plume that propagated southward strongly modified the composition of the atmosphere and the rain that fell in the days following the eruption throughout the south-eastern sector of Sicily. The strong anomaly in fluoride deposition observed in January at all the sites of Priolo Gargallo AERCA, and also at Palazzolo Acreide, but absent at the Milazzo AERCA sites, is related to this episode [21] (Figure 11c and Figure 12c).

Monthly calcium depositions were fairly homogeneous throughout the study period, although higher median values were measured at the Priolo Gargallo AERCA sites (118 mg m^−2^) than at the Milazzo AERCA sites (100 mg m^−2^). Calcium derived mainly from the local geogenic source in both study areas, with a strong contribution also represented by the dust of Saharan origin that frequently reached the Mediterranean basin. The study area of the Priolo Gargallo AERCA, as well as the Palazzolo Acreide area, were more exposed than the Milazzo AERCA area to the sirocco winds, responsible for the transport of Saharan dust in the Mediterranean area, and this could explain, together with the greater extent of carbonate outcrops in this area, the higher median deposition values in the former rather than in the latter study area. The highest calcium deposition values were recorded during November 2018 both in the AERCA of Milazzo (up to 393 mg m^−2^) and especially in the AERCA of Priolo Gargallo (up to 649 mg m^−2^) and at Palazzolo Acreide (463 mg m^−2^) and could be related to Saharan dust carried by the sirocco winds which affected the Mediterranean between the 20th and 25th of the same month (Figure 11d and Figure 12d).

A comparison with atmospheric depositions reported for rainwater sampled by other authors in different areas of the Mediterranean basin highlighted some peculiar features of the rainwater studied in this research. The fluoride deposition values were lower than those recorded by Calabrese et al. 2011 [42] in the Etna area. Fluoride was one of the gaseous species emitted by Etna in the form of HF, therefore its concentrations in the atmosphere decreased on moving away from the emission vents (Figure 13). Median total depositions of chloride were higher than those recorded by Al-Momani et al. 1995 [53] in the eastern Mediterranean basin, by Pieri et al. 2010 [64] in northern Italy, by Calabrese et al. 2011 [42] in Etna area, by Morales-Baquero et al. 2013 [65] in southern Spain, by Castillo et al. 2017 [66] at Montesy–Barcelona, and by Cerro et al. 2020 [67] at Mallorca Island. The high value of annual chloride deposition in our work was explained by the close proximity of the monitoring sites to the coast. The lowest values were recorded at Palazzolo Acreide (8.49 ± 0.64 g m^−2^ yr^−1^) and Melilli (9.84 ± 0.65 g m^−2^ yr^−1^), i.e., at the two sites located at a higher elevation and greater distance from the sea. It is not surprising, therefore, that the chloride deposition value most similar to those measured in the present research belongs to the work of Al-Momani et al. 1995 [53], carried out at a location close to the sea (Figure 13). Median annual deposition of nitrate, potassium and magnesium were in the same range as found in the Mediterranean region (Figure 13). Median total deposition of sulphate was higher than those reported for other Mediterranean areas (northern Italy, southern Spain, Montesy–Barcelona, and Mallorca Island), similar to that reported by Al-Momani et al. 1995 [53], but lower than that reported by Calabrese et al. 2011 [42] for the Etna area, the volcano being the major point source of gaseous sulphur species for the Mediterranean area (Figure 13). Annual depositions of sodium, having a predominantly marine origin, and calcium, having a geogenic origin, were similar to those reported by Al-Momani et al. 1995 [53] for the coastal site in Turkey and higher than those measured in other areas of the Mediterranean (Figure 13).

## 5. Conclusions

A study of the inorganic chemical composition of rainwater was carried out for almost one year in two areas around two big industrial hubs in Eastern Sicily. The main sources influencing the rainwater composition in the study areas include marine, terrigenous (crustal), anthropogenic, and volcanic sources. Rainwater had low mineralization, with low Electrical Conductivity values (inversely correlated with the rain amounts), and pH, usually higher than 5.6, up to 8.3, due to high concentrations of mostly geogenic Ca^2+^ which played an important buffering role of the natural rainwater acidity of the study areas. Terrigenous dust came from both local limestone outcrops and the Saharan region. The chemical composition of the rainwater was strongly influenced by sea spray, showing a high load of Na^+^, Cl^−^, and Mg^2+^ in all the samples from the coastal sites, while the concentrations of the same elements decreased when moving away from the sea. Another important source for the chemical species was represented by anthropogenic activities, the study areas being near to two of Europe’s most important industrial and fossil fuel refining hubs, responsible for significant emissions of anthropogenic gaseous species such as CO_2_, SO_x_, and NO_x_. High concentrations of NO_3_^−^ (similar to those reported by other authors for different Mediterranean sites) and SO_4_^2−^ (lower than those reported for the nearby volcanic area of Mt. Etna, but higher than those reported for other Mediterranean study sites) have been measured. The positive correlation between nitrate and nss-SO_4_^2−^ (R^2^ = 0.68) corroborates the hypothesis that both can be attributed to the same anthropogenic sources, i.e., fossil fuel combustion. Fluoride shows low deposition values in all the monitoring sites, even if higher median values were measured in the samples of the Milazzo AERCA than in that of the Priolo Gargallo AERCA, due to the presence of some local anthropogenic sources such as cement production. A major anomaly in fluorine deposition of volcanic origin was observed at the Priolo Gargallo and Palazzo Acreide sites following the eruption of Etna in late December 2018.

The scientific evidence produced by the present research highlights the multiplicity of sources (both natural and anthropogenic), which may contribute to the chemical composition of rainwater, and the way to discriminate them. The geographical location of Sicily in a central position in the Mediterranean basin and the proximity of the study areas to major industrial hubs have proved to be factors of great importance in the study of the chemical composition of atmospheric deposition. The role of industrial processes in conditioning the chemical composition of the atmosphere and the epidemiological evidence in the study areas call for more in-depth environmental studies in the same areas.

## Figures and Tables

**Figure 1 ijerph-20-03898-f001:**
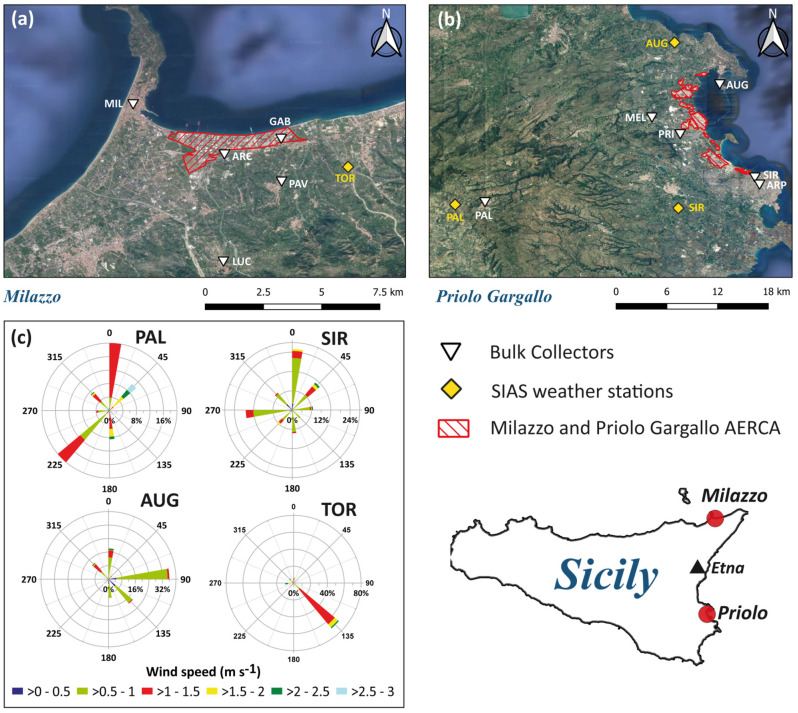
Milazzo AERCA (**a**) and Priolo Gargallo AERCA (**b**) (red dashed areas); location of the SIAS weather station (yellow squares), and monitoring sites (white triangles). Box (**c**) shows the wind speeds and directions recorded at 2 m above the ground at SIAS weather stations. Base map: Google Earth.

**Figure 2 ijerph-20-03898-f002:**
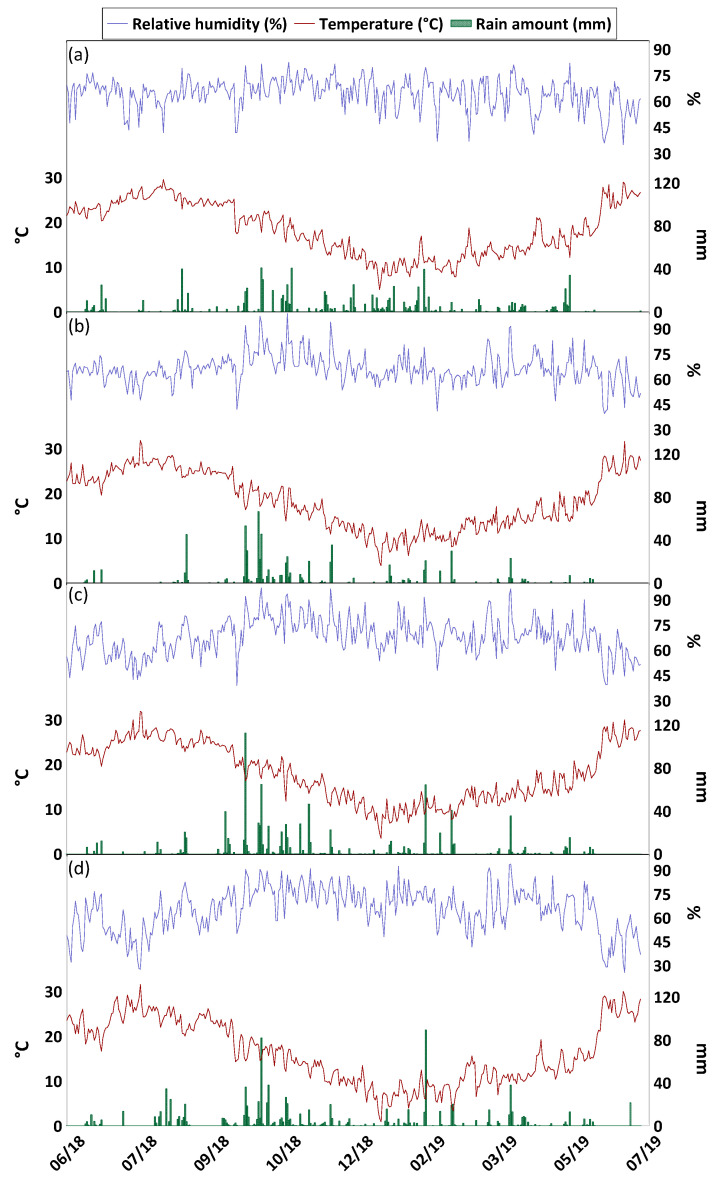
Temperature (°C), relative humidity (%), and rain amount (mm) at the SIAS automatic weather station of Milazzo (**a**), Augusta (**b**), Siracusa (**c**), and Palazzolo Acreide (**d**), during the study period.

**Figure 3 ijerph-20-03898-f003:**
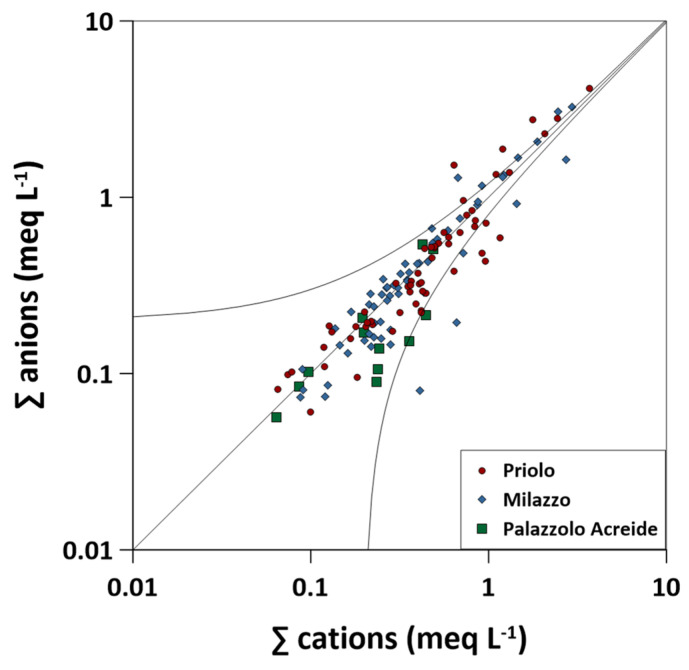
Sum of anions vs. sum of cations binary diagram. ∑ anions = Cl^−^ + NO_3_^−^ + SO_4_^2−^ + HCO_3_^−^ + F^−^ and ∑ cations = Ca^2+^ + Na^+^ + Mg^2+^ + K^+^. The black line (1:1) represents the charge balance between anions and cations. The area between the two black curves represents the acceptable balance error following the standard analytical methods (APHA, AWWA, WEF, 2005 [51]).

**Figure 4 ijerph-20-03898-f004:**
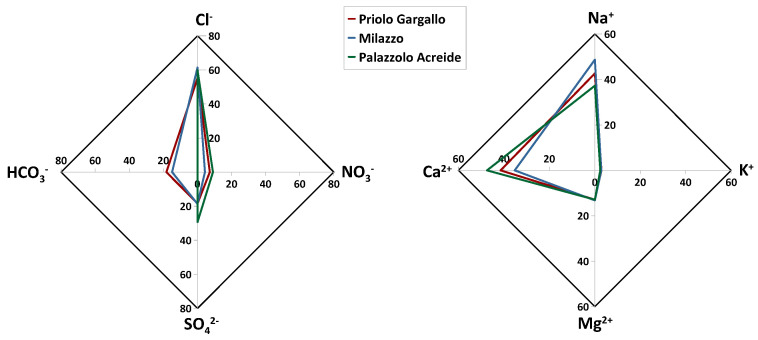
Major ions’ relative abundances (%), on an equivalent basis, in the study areas of Priolo Gargallo (red), Milazzo (blue), and Palazzolo Acreide (green).

**Figure 5 ijerph-20-03898-f005:**
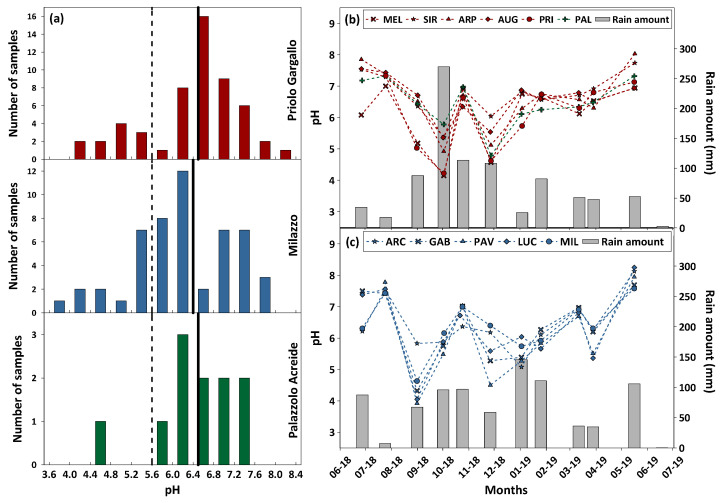
(**a**) The pH values of bulk deposition samples collected from June 2018 to June 2019 at Priolo Gargallo, Milazzo, and Palazzolo Acreide. The dashed line indicates the theoretical pH value expected for unpolluted cloud water at equilibrium with 410 ppm of atmospheric CO_2_. The solid lines indicate the median pH values in the study areas. On the right side, seasonal variability of the pH values and average rain amount at Priolo Gargallo (**b**) and Milazzo (**c**) study areas.

**Figure 6 ijerph-20-03898-f006:**
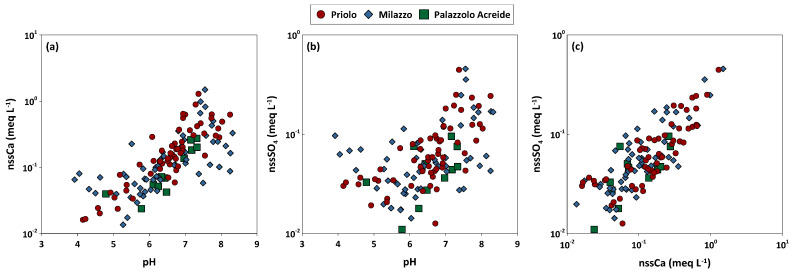
Correlation between pH values and nssCa^2+^ (meq L^−1^) (**a**), pH and nnsSO_4_^2−^ (meq L^−1^) (**b**), and between nssCa^2+^ (meq L^−1^) and nnsSO_4_^2−^ (meq L^−1^) (**c**), in all study areas.

**Figure 7 ijerph-20-03898-f007:**
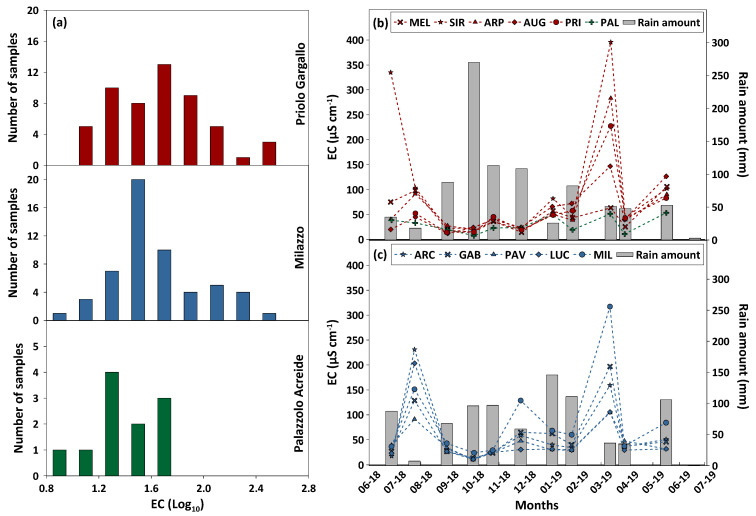
(**a**) The EC distribution (Log_10_) of rainwater samples that were collected from June 2018 to June 2019 at Priolo Gargallo, Milazzo, and Palazzolo Acreide study areas, respectively. On the right side, seasonal variability of the EC values and average rain amount at Priolo Gargallo (**b**) and Milazzo (**c**) study areas.

**Figure 8 ijerph-20-03898-f008:**
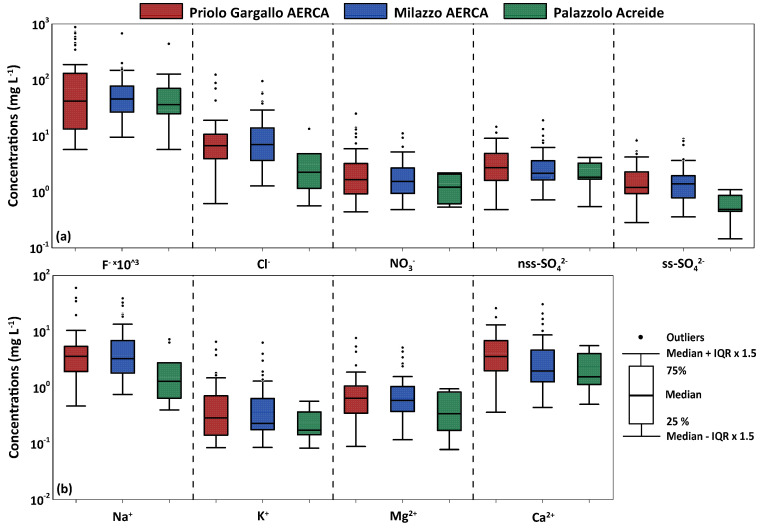
Major anions (**a**) and cations (**b**) concentrations (mg L^−1^) for Priolo Gargallo AERCA, Milazzo AERCA, and Palazzolo Acreide background area.

**Figure 9 ijerph-20-03898-f009:**
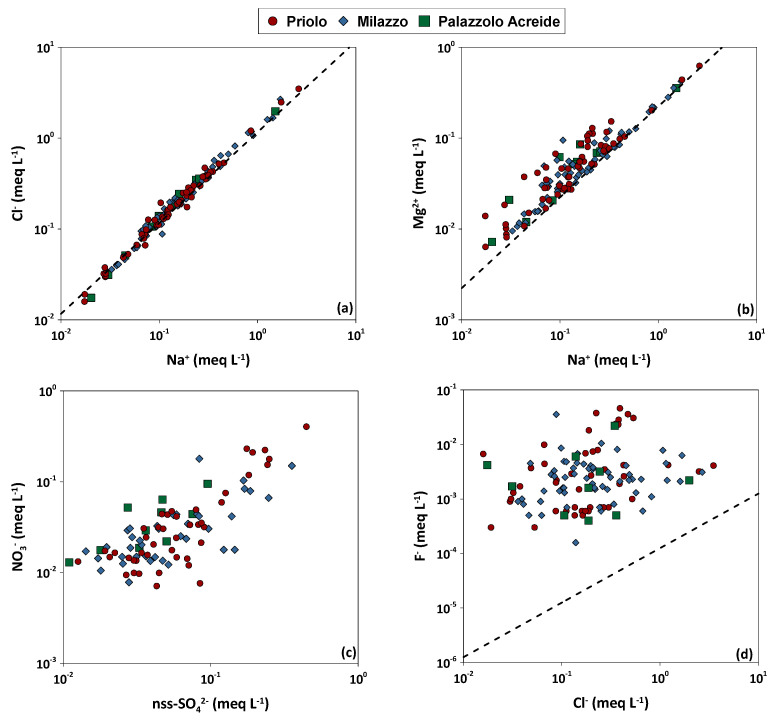
Correlations, in (meq L^−1^), in Priolo, Milazzo, and Palazzolo Acreide study areas, between Na^+^ and Cl^−^ (**a**), Na^+^ and Mg^2+^ (**b**), nss-SO_4_^2−^ and NO_3_^−^ (**c**), Cl^−^ and F^−^ (**d**). In (**a**,**b**,**d**) the dotted line represents the ratio of ions in seawater (Keene, 1986 [9]).

**Figure 10 ijerph-20-03898-f010:**
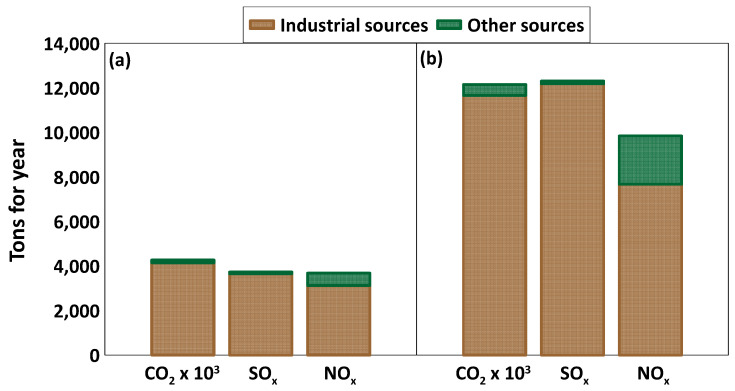
Gaseous species (CO_2_, SO_x_, NO_x_) emissions and sources at Milazzo AERCA (**a**) and Priolo Gargallo AERCA (**b**) (ARPA, 2022, [62]).

**Figure 11 ijerph-20-03898-f011:**
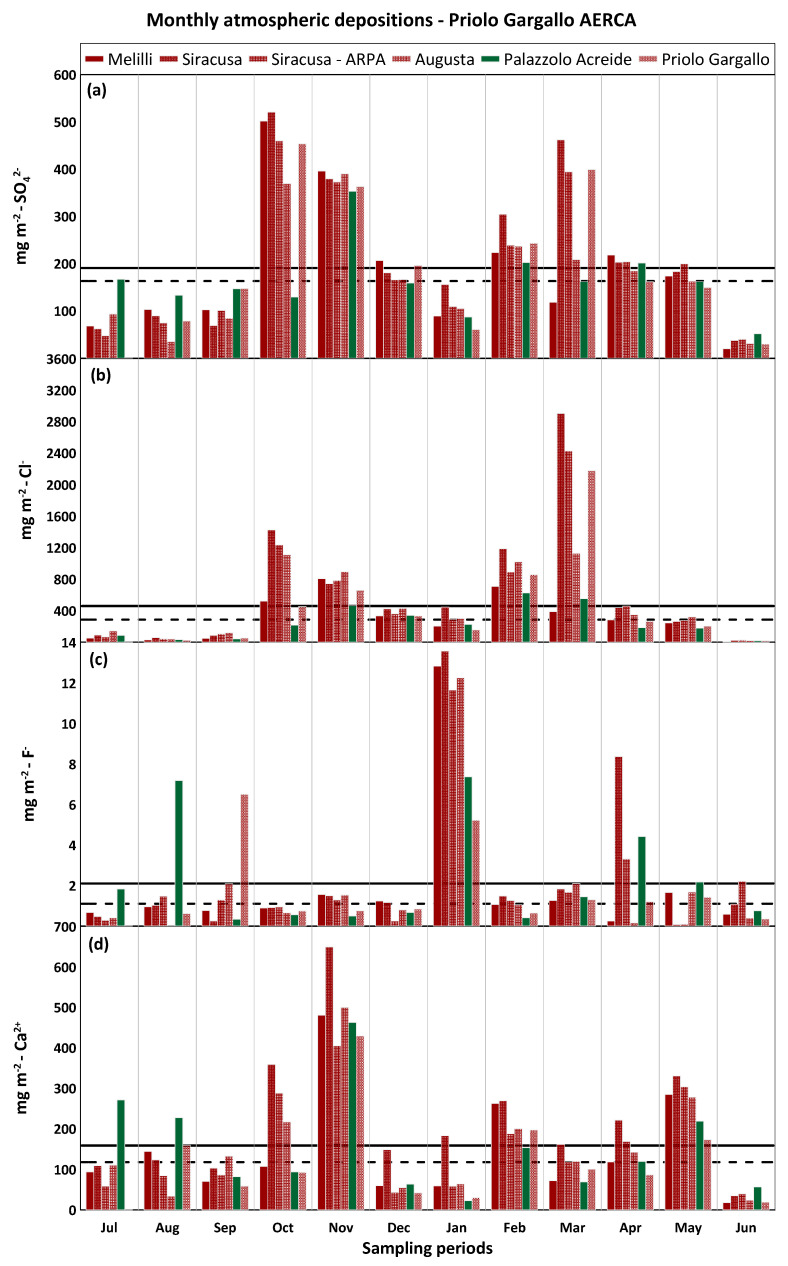
Monthly atmospheric depositions (mg m^−2^) for sulphate (**a**), chloride (**b**), fluoride (**c**), and calcium (**d**) at Priolo Gargallo AERCA. The solid black lines are mean values; the dashed lines are median values.

**Figure 12 ijerph-20-03898-f012:**
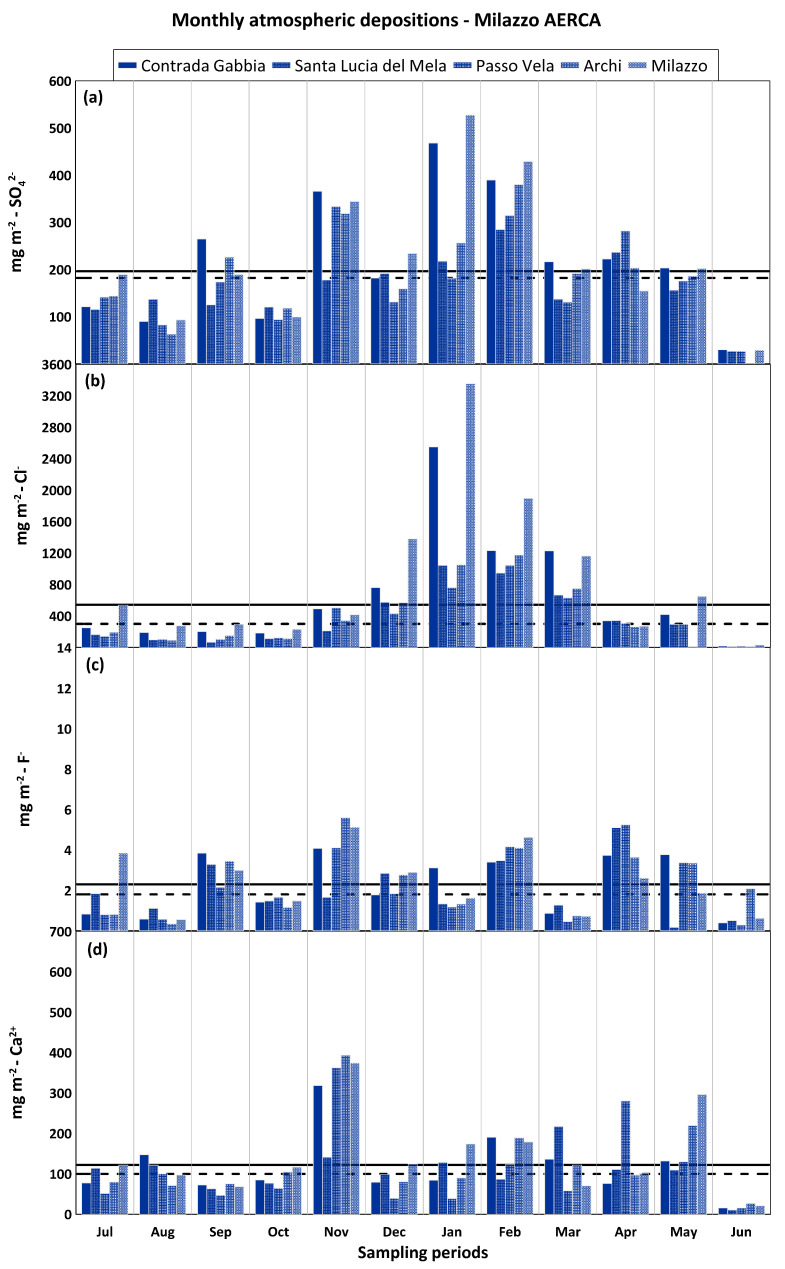
Monthly atmospheric depositions (mg m^−2^) for sulphate (**a**), chloride (**b**), fluoride (**c**), and calcium (**d**) at Milazzo AERCA. The solid black lines are average values; the dashed lines are median values.

**Figure 13 ijerph-20-03898-f013:**
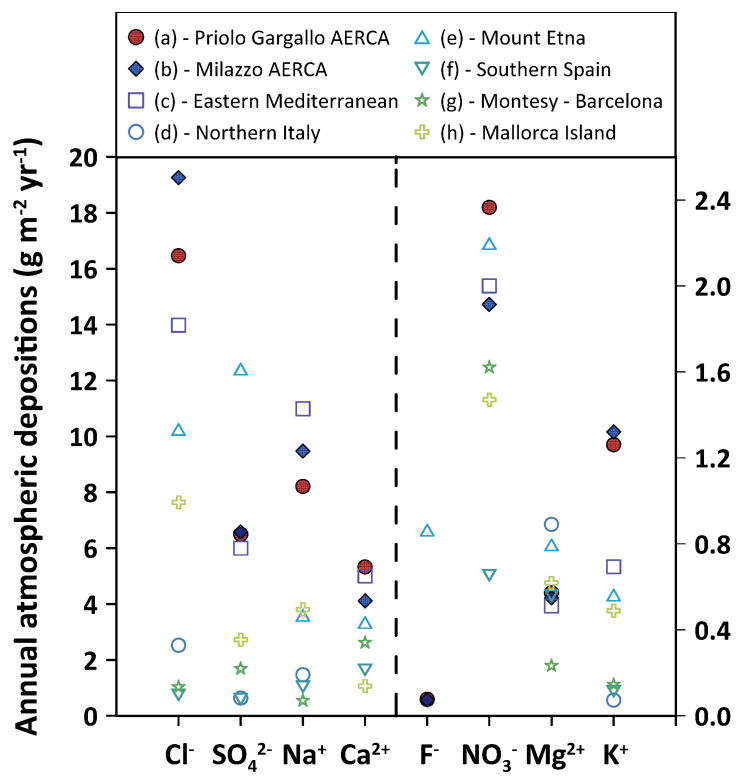
Annual atmospheric depositions (g m^−2^ yr^−1^). (**a**) this study—Priolo Gargallo AERCA; (**b**) this study—Milazzo AERCA; (**c**) Izmir, Turkey (Al-Momani et al. 1995 [53]); (**d**) northern Italy (Pieri et al. 2010 [64]); (**e**) Mount Etna (Calabrese et al. 2011 [42]); (**f**) southern Spain (Morales-Baquero et al. 2013 [65]); (**g**) Montesy–Barcelona (Castillo et al. 2017 [66]); (**h**) Mallorca Island (Cerro et al. 2020 [67]). For chloride, sulphate, sodium, and calcium see the axis on the left; for fluoride, nitrate, magnesium, and potassium see the axis on the right.

**Table 1 ijerph-20-03898-t001:** The collector network and information of monitoring sites.

Locality	ID	AERCA	Type of Site	Position UTM WGS84	Altitude (m a.s.l.)	Distance from Sea (km)
Contrada Gabbia	GAB	Milazzo	Industrial	33S 527148 4228670	15	0.43
Archi	ARC	Milazzo	Industrial	33S 524760 4227710	34	1.05
Pace del Mela	PAV	Milazzo	Urban	33S 527271 4226464	107	2.55
S. Lucia del Mela	LUC	Milazzo	Urban	33S 524222 4222617	150	6.11
Milazzo	MIL	Milazzo	Urban	33S 520799 4230495	20	0.36
Siracusa	SIR	Priolo Gargallo	Urban	33S 523613 4106275	63	0.44
Siracusa	ARP	Priolo Gargallo	Urban	33S 524666 4104347	76	1.95
Melilli	MEL	Priolo Gargallo	Urban	33S 511433 4115107	249	5.15
Augusta	AUG	Priolo Gargallo	Industrial	33S 519388 4120387	28	0.10
Priolo Gargallo	PRI	Priolo Gargallo	Industrial	33S 514767 4112657	94	2.97
Palazzolo Acreide	PAL	-	Semi-Urban	33S 491728 4102587	642	28.2

**Table 2 ijerph-20-03898-t002:** Statistical information on the measured pH, electrical conductivity (μS cm^−1^), and major ion concentration (meq L^−1^) values, for both the study areas. VWM = volume-weighted mean. ss% = percent of sea salt contribution calculated on the VWMs. nss% = percent of non-sea salt contribution calculated on the VWMs.

Site		pH	EC	F^−^	Cl^−^	NO_3_^−^	SO_4_^2−^	Na^+^	K^+^	Mg^2+^	Ca^2+^
			μS cm^−1^	meq L^−1^	meq L^−1^	meq L^−1^	meq L^−1^	meq L^−1^	meq L^−1^	meq L^−1^	meq L^−1^
Melilli	Min	4.15	15	<0.003	0.03	0.014	0.04	0.03	0.002	0.01	0.03
	Max	7.95	106	0.0461	0.43	0.119	0.19	0.30	0.037	0.11	0.63
	VWM		30		0.13	0.022	0.06	0.09	0.006	0.03	0.11
	ss (%)				88	0	20	100	33	68	4
	nss (%)				12	100	80	0	67	32	96
Siracusa	Min	5.37	21	<0.003	0.05	<0.005	0.03	0.05	0.003	0.02	0.08
	Max	7.74	396	0.0306	3.49	0.22	0.41	2.62	0.097	0.62	0.90
	VWM		58		0.32	0.014	0.07	0.25	0.009	0.07	0.17
	ss (%)				90	0	42	100	60	83	6
	nss (%)				10	100	58	0	40	17	94
Siracusa-Arpa	Min	4.92	19	<0.003	0.09	<0.005	0.03	0.07	0.002	0.02	0.05
	Max	8.25	283	0.0283	2.49	0.403	0.48	1.74	0.122	0.44	1.30
	VWM		44		0.28	0.018	0.07	0.21	0.007	0.06	0.12
	ss (%)				86	0	37	100	63	79	8
	nss (%)				14	100	63	0	37	21	92
Augusta	Min	5.36	12	<0.003	0.03	<0.005	0.02	0.03	0.002	0.01	0.06
	Max	7.92	147	0.0359	1.21	0.231	0.22	0.86	0.038	0.20	0.66
	VWM		40		0.23	0.012	0.06	0.18	0.006	0.05	0.13
	ss (%)				90	0	37	100	66	80	6
	nss (%)				10	100	63	0	34	20	94
Palazzolo Acreide	Min	4.79	7	<0.003	0.02	<0.005	0.01	0.02	0.002	0.01	0.03
	Max	7.33	53	0.0232	0.39	0.035	0.11	0.32	0.015	0.08	0.28
	VWM		25		0.09	0.016	0.04	0.08	0.005	0.03	0.10
	ss (%)				97	0	21	100	37	66	3
	nss (%)				3	100	79	0	63	34	97
Priolo Gargallo	Min	4.62	14	<0.003	0.02	<0.005	0.04	0.02	0.020	0.01	0.02
	Max	7.33	227	0.0220	1.98	0.095	0.27	1.52	0.168	0.35	0.42
	VWM		41		0.23	0.022	0.07	0.18	0.007	0.05	0.09
	ss (%)				89	0	31	100	54	81	8
	nss (%)				11	100	69	0	46	19	92
Contrada Gabbia	Min	4.32	27	<0.003	0.06	<0.005	0.02	0.06	0.002	0.02	0.03
	Max	7.78	197	0.0068	1.60	0.079	0.21	1.26	0.030	0.28	0.68
	VWM		47		0.31	0.014	0.07	0.23	0.007	0.06	0.09
	ss (%)				84	0	38	100	75	90	11
	nss (%)				16	100	62	0	25	10	89
S. Lucia del Mela	Min	4.05	11	<0.003	0.04	<0.005	0.03	0.03	0.030	0.01	0.04
	Max	8.26	203	0.0086	0.64	0.150	0.18	0.42	0.162	0.12	0.84
	VWM		33		0.18	0.019	0.05	0.13	0.007	0.04	0.09
	ss (%)				86	0	30	100	43	85	6
	nss (%)				14	100	70	0	57	15	94
Passo Vela	Min	3.92	10	<0.003	0.04	<0.005	0.02	0.04	0.003	0.01	0.02
	Max	7.95	105	0.0049	0.82	0.179	0.18	0.59	0.033	0.13	0.51
	VWM		34		0.17	0.020	0.06	0.14	0.006	0.04	0.08
	ss (%)				91	0	29	100	51	80	8
	nss (%)				9	100	71	0	49	20	92
Archi	Min	5.08	13	<0.003	0.04	<0.005	0.03	0.04	0.002	0.01	0.05
	Max	8.13	159	0.0356	1.18	0.072	0.56	0.90	0.077	0.22	1.53
	VWM		38		0.20	0.008	0.07	0.18	0.007	0.05	0.12
	ss (%)				100	0	30	100	54	83	7
	nss (%)				0	100	70	0	46	17	93
Milazzo	Min	4.63	24	<0.003	0.11	<0.005	0.03	0.07	0.003	0.03	0.06
	Max	8.32	317	0.0105	2.68	0.103	0.42	1.70	0.103	0.42	1.05
	VWM		64		0.47	0.014	0.08	0.35	0.009	0.09	0.13
	ss (%)				86	0	50	100	90	88	12
	nss (%)				14	100	50	0	10	12	88

**Table 3 ijerph-20-03898-t003:** The correlation coefficient between major ions for the Milazzo (a) and the Priolo Gargallo (b) bulk depositions. In bold the R^2^ values ≥ than 0.60.

(a)	F^−^	Cl^−^	NO_3_^−^	SO_4_^2−^	Na^+^	K^+^	Mg^2+^	Ca^2+^	nss-SO_4_^2−^
F^−^		<0.10	0.19	0.45	<0.10	0.09	<0.10	0.13	**0.60**
Cl^−^	<0.10		<0.10	0.43	**0.98**	0.22	**0.95**	0.20	0.11
NO_3_^−^	0.19	<0.10		0.30	<0.10	0.37	<0.10	0.22	0.47
SO_4_^2−^	0.45	0.43	0.30		0.49	**0.74**	0.57	**0.81**	**0.86**
Na^+^	<0.10	**0.98**	<0.10	0.49		0.27	**0.97**	0.27	0.15
K^+^	0.09	0.22	0.37	**0.74**	0.27		0.36	**0.60**	**0.71**
Mg^2+^	<0.10	**0.95**	<0.10	0.57	**0.97**	0.36		0.35	0.21
Ca^2+^	0.13	0.20	0.22	**0.81**	0.27	**0.60**	0.35		**0.78**
nss-SO_4_^2−^	**0.60**	0.11	0.47	**0.86**	0.15	**0.71**	0.21	**0.78**	
**(b)**	**F^−^**	**Cl^−^**	**NO_3_^−^**	**SO_4_^2−^**	**Na^+^**	**K^+^**	**Mg^2+^**	**Ca^2+^**	**nss-SO_4_^2−^**
F^−^		<0.10	0.11	0.15	<0.10	0.09	<0.10	0.09	0.19
Cl^−^	<0.10		<0.10	0.37	**0.99**	0.15	**0.95**	<0.10	<0.10
NO_3_^−^	0.11	<0.10		**0.79**	<0.10	0.21	0.11	**0.75**	**0.89**
SO_4_^2−^	0.15	0.37	**0.79**		0.39	0.44	0.52	0.59	**0.71**
Na^+^	<0.10	**0.99**	<0.10	0.39		0.16	**0.96**	<0.10	<0.10
K^+^	0.09	0.15	0.21	0.44	0.16		0.23	0.17	0.33
Mg^2+^	<0.10	**0.95**	0.11	0.52	**0.96**	0.23		0.07	0.06
Ca^2+^	0.09	<0.10	**0.75**	0.59	<0.10	0.17	0.07		**0.82**
nss-SO_4_^2−^	0.19	<0.10	**0.89**	**0.71**	<0.10	0.33	0.06	**0.82**	

**Table 4 ijerph-20-03898-t004:** Calculated annual atmospheric depositions (g m^−2^ yr^−1^).

	Atmospheric Depositions (g m^−2^ yr^−1^)	F^−^	Cl^−^	NO_3_^−^	SO_4_^2−^	Na^+^	K^+^	Mg^2+^	Ca^2+^	Rainfall mm
Priolo Gargallo AERCA	Melilli	0.08 ± 0.011	9.84 ± 0.65	3.06 ± 0.18	6.11 ± 0.37	4.81 ± 0.31	0.55 ± 0.03	0.85 ± 0.05	4.92 ± 0.37	805
	Siracusa	0.09 ± 0.014	25.35 ± 3.30	1.94 ± 0.24	7.77 ± 0.55	12.74 ± 1.61	0.78 ± 0.10	1.84 ± 0.20	7.58 ± 0.44	816
	Siracusa-Arpa	0.08 ± 0.011	21.68 ± 2.75	2.38 ± 0.27	7.08 ± 0.47	10.44 ± 1.26	0.61 ± 0.05	1.57 ± 0.17	5.29 ± 0.38	801
	Augusta	0.08 ± 0.011	16.91 ± 1.36	1.50 ± 0.17	5.79 ± 0.31	8.48 ± 0.66	0.47 ± 0.02	1.27 ± 0.09	5.32 ± 0.37	744
	Priolo Gargallo	0.06 ± 0.008	16.51 ± 2.53	2.72 ± 0.17	6.74 ± 0.48	8.22 ± 1.26	0.56 ± 0.05	1.22 ± 0.15	3.80 ± 0.27	733
	Median	0.08 ± 0.011	16.91 ± 2.53	2.38 ± 0.18	6.74 ± 0.47	8.48 ± 1.26	0.56 ± 0.05	1.27 ± 0.15	5.29 ± 0.37	801
Milazzo AERCA	Contrada Gabbia	0.08 ± 0.005	24.30 ± 2.49	1.88 ± 0.17	7.68 ± 0.42	11.40 ± 1.07	0.57 ± 0.04	1.52 ± 0.12	3.97 ± 0.19	798
	Santa Lucia del Mela	0.06 ± 0.003	13.42 ± 1.14	2.45 ± 0.13	5.36 ± 0.19	6.42 ± 0.55	0.55 ± 0.03	0.91 ± 0.07	3.80 ± 0.22	763
	Passo Vela	0.07 ± 0.005	12.84 ± 0.93	2.60 ± 0.29	5.65 ± 0.22	6.54 ± 0.47	0.47 ± 0.02	0.98 ± 0.06	3.30 ± 0.21	755
	Archi	0.08 ± 0.005	13.49 ± 1.25	0.92 ± 0.12	6.40 ± 0.30	7.59 ± 0.60	0.52 ± 0.03	1.10 ± 0.08	4.43 ± 0.27	680
	Milazzo	0.08 ± 0.003	32.25 ± 3.18	1.72 ± 0.16	7.81 ± 0.45	15.42 ± 1.52	0.64 ± 0.04	2.10 ± 0.19	5.05 ± 0.34	699
	Median	0.08 ± 0.005	13.49 ± 1.25	1.88 ± 0.16	6.40 ± 0.30	7.59 ± 0.60	0.55 ± 0.03	1.10 ± 0.09	3.97 ± 0.22	755
	Palazzolo Acreide	0.08 ± 0.008	8.49 ± 0.64	2.58 ± 0.14	5.47 ± 0.18	4.57 ± 0.34	0.45 ± 0.01	0.82 ± 0.05	5.06 ± 0.32	937

## Data Availability

All data has been published in a public repository, https://doi.org/10.26022/IEDA/112778.

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
