# Peer review of "Atmospheric Deposition around the Industrial Areas of Milazzo and Priolo Gargallo (Sicily–Italy)—Part A: Major Ions"

_ijerph, 2023, doi:10.3390/ijerph20053898_

Round 1
Reviewer 1 Report
The manuscript is well-written and well-organized in its presentation. It can be published. However, there are some small revisions to be made:
1. Caption in Figure 1 is not complete. Please, explain the a), b), c) parts in details.
2. Caption in Figure 6 shall be fixed indicating a), b), and c).
3. Double - check if all references are reported according to the journal's policies (e.g. lines 589-592).
4. Any standards were used in the analitycal determinations?
Author Response
Reply to the Review 1:
- Caption in Figure 1 is not complete. Please, explain the a), b), c) parts in details.
Thank you for the observation. A detailed explanation has been added to the caption.
- Caption in Figure 6 shall be fixed indicating a), b), and c).
Done
- Double - check if all references are reported according to the journal's policies (e.g. lines 589-592).
A review of the bibliography was conducted and some changes were made. They can be observed in the "Track Changes" version. Thank you for your observation.
- Any standards were used in the analytical determinations?
Required details were added in Materials and Methods (see lines 229-234 in the “Track Changes” version).
Author Response
Reply to Review 2:
Abstract
Lines 24-27 Sentence too long. Please rephrase.
The sentence has been rewritten as “Concentrations of major ionic species followed the sequence Cl- > Na+ > SO42- ≃ HCO3- > ≃ Ca2+ > NO3- > Mg2+ > K+ > F-. High loads of Na+ and Cl- (with a calculated R2 = 0.99) reflected proximity to the sea. Calcium, potassium, and non-sea-salt magnesium had a prevalent crustal origin.” (See lines 24-27 in the “Track Changes” version).
Lines 24.25 “The study areas were characterised by large oil reefing plants and other industrial hubs that release large amounts of gaseous species capable of affecting atmospheric chemical composition” should be taken to the second sentence of the abstract.
Thank you for the advice. The sentence has been moved in the first part of the abstract. (See lines 15-18 in the “Track Changes” version).
Introduction
Line 35 Correct to (… and remove through different physicochemical mechanism).
The sentence has been changed as suggested.
Line 41 correct to …… salts and crustal dust which is one of the prominent aerosols in the atmosphere [3,4]. Remove the parenthesis leaving only the square bracket containing the citations and end the sentence. Start new sentence with volcanic emission, biogenic……
The sentence has been changed as suggested.
Line 89 Correct to “ The European Environmental Agency reported the existence of hundreds ……
Take lines 89-94 to the introduction.
The sentence has been changed as suggested and moved to the introduction (See lines 86-91 in the “Track Changes” version).
2 Study area and climatic setting
Begin with The study area is located in Sicily….
Done
Take lines 114 to 121 to the concluding part of the introduction as statement of problem before ending with the introduction with aims and objectives.
Thank you for the suggestion. The sentences have been moved to the concluding part of the introduction. (See lines 92-99 in the “Track Changes” version).
Results and discussion
Lines 206 to 232 be transferred to Materials and Methods as last paragraph captioned as “ Analysis of Chemical Data”. All equations should come under this sub heading . result and discussion sentence begins with 4.1 Chemical Composition of rain water and the atmosphere.
Lines 310, 317 and all equations should be taken to Analysis of Chemical data under materials and methods.
Thank you for the observation. All the equations have been moved to a new paragraph named “Analysis of Chemical Data”. (See lines 243-294 in the “Track Changes” version).
Lines 388-454 Reduce the length of this paragraph by discussing only major ions that are extremely high or low. Intermediate levels major ions discussion not necessary.
Thank you for the observation. The paragraph has been rewritten as suggested. (See lines 368-387 in the “Track Changes” version).
My concern with this study is that you are not comparing your established water and atmospheric pollutants with any known world standards. Why?
The comparison with known world standards was done. The list of standards used has been included in the Materials and Methods section (See lines 229-234 in the “Track Changes” version).
Reviewer 3 Report
The manuscript is devoted to the study of the cationic and anionic composition of atmospheric precipitation in two cities on Sicily island. Precipitation samples were collected within one year and analyzed using common analytical methods. The authors linked variations in the content of main cations and anions in atmospheric precipitation with the influence of both natural and anthropogenic factors.
Remarks:
- the authors mention HCO3- (Line 254, Fig. 4), however, the chapter "Materials and methods" does not describe the procedure for the analytical determination of this parameter in samples;
- when discussing the results of the work, the authors use different statistical parameters as the average value (arithmetic mean, geometric mean, median), however, taking into account the database, most likely in this case we are dealing with a heterogeneous database, so it is better to use the median / geometric mean as the average.
Author Response
Reply to Review 3:
The manuscript is devoted to the study of the cationic and anionic composition of atmospheric precipitation in two cities on Sicily island. Precipitation samples were collected within one year and analyzed using common analytical methods. The authors linked variations in the content of main cations and anions in atmospheric precipitation with the influence of both natural and anthropogenic factors.
Remarks:
- the authors mention HCO3- (Line 254, Fig. 4), however, the chapter "Materials and methods" does not describe the procedure for the analytical determination of this parameter in samples.
Thanks for the comment; total alkalinity corresponds to the HCO3- (plus CO32- if present). A sentence was added to better explain in Materials and Methods (see line 222 in the “Track Changes” version).
- when discussing the results of the work, the authors use different statistical parameters as the average value (arithmetic mean, geometric mean, median), however, taking into account the database, most likely in this case we are dealing with a heterogeneous database, so it is better to use the median / geometric mean as the average.
Thank you for the observation. We agree that the database is heterogeneous. Where useful, values of arithmetic averages have been replaced with medians.